



# Technical note: Improving the AWAT filter with interpolation schemes for advanced processing of high resolution data

Andre Peters[1], Thomas Nehls[1], Gerd Wessolek[1]

[1]Institut für Ökologie, Technische Universität Berlin, Berlin, 10587, Germany

*Correspondence to*: A. Peters (andre.peters@tu-berlin.de)

**Abstract.** Weighing lysimeters with appropriate data filtering yield the most precise and unbiased information for precipitation (P) and evapotranspiration (ET). A recently introduced filter scheme for such data is the AWAT (Adaptive Window and Adaptive Threshold) filter [Peters, A., Nehls, T., Schonsky, H., and Wessolek, G.: Separating precipitation and evapotranspiration from noise – a new filter routine for high-resolution lysimeter

data, Hydrol. Earth Syst. Sci., 18, 1189–1198, doi:10.5194/hess-18-1189-2014, 2014]. The filter applies an adaptive threshold to separate significant from insignificant mass changes, guaranteeing that P and ET are not overestimated, and uses a step interpolation between the significant mass changes. In this contribution we show that the step interpolation scheme, which reflects the resolution of the measuring system, can lead to unrealistic prediction of P and ET, especially if they are required in high temporal resolution. We introduce linear and spline

interpolation schemes to overcome these problems. To guarantee that medium to strong precipitation events abruptly following low or zero fluxes are not smoothed in an unfavourable way, a simple heuristic selection criterion is used, which attributes such precipitations to the step interpolation. The three interpolation schemes (step, linear and spline) are tested and compared using a data set from a grass-reference lysimeter with one minute resolution, ranging from 1 January to 5 August 2014. The selected output resolutions for P and ET

prediction are one day, one hour and 10 minutes. As expected, the step scheme yielded reasonable flux rates only for a resolution of one day, whereas the other two schemes are well able to yield reasonable results for any resolution. The spline scheme returned slightly better results than the linear scheme concerning the differences between filtered values and raw data. Moreover, this scheme allows continuous differentiability of filtered data so that any output resolution for the fluxes is sound. Since computational burden is not problematic for any of the

interpolation schemes, we suggest to use always the spline scheme.

## 1 Introduction

Precipitation (P [L T⁻¹]) and evapotranspiration (ET [L T⁻¹]) have to be precisely known to answer many questions regarding water, solute and energy fluxes in the soil-plant atmosphere continuum. In several simulation studies, the precise values for P and ET are required only as daily averages (e.g. Schelle et al., 2012). However, in other





cases the diurnal course of P and ET must be known, e.g. if root water uptake shall be simulated with a physically based model (Javaux et al., 2008; Couvreur et al., 2012) or macro-pore flow due to heavy but short precipitation events shall be simulated under realistic conditions (Malone et al., 2004; McGrath et al., 2008).

Today, weighing lysimeter measurements with a high mass and temporal resolution yield the most precise values

for both P and ET. This is since systematic as well as random errors are largely eliminated; the former due to their installation height exactly at ground surface and the latter due to the relatively large size in comparison to other devices. The high temporal resolution of the measurement is required to distinguish between P and ET, which might follow each other even in small time intervals.

The mass resolution of the lysimeter can be as high as 0.01 mm for modern weighing systems (von Unold and

Fank, 2008) and can be even used for dew fall measurements (Meissner et al., 2007). With such high resolutions, small disturbances, e.g. due to wind, are visible in the data as noise (Nolz et al., 2013) and must be eliminated before the data can be interpreted (Fank, 2013; Schrader et al., 2013; Peters et al., 2014). Moreover, the disturbance, and thus the accuracy, of the system depends on wind speed and is therefore not constant but time variable. After elimination of the measurement noise with appropriate filter routines each increase in system

mass is interpreted as precipitation and each decrease as evapotranspiration.

As already suggested by Fank (2013) and Schrader et al. (2013) such filter routines can be carried out in two steps. First a smoothing routine (for example a simple moving average) with a certain window width $w$ [T] is applied and second all changes of the smoothed data smaller than a predefined threshold value $\delta$ [L] are discarded. The second step is mandatory to avoid that small changes of the smoothed data are interpreted as P

and ET. Schrader et al. (2013) showed that there are no "ideal" values for $w$ or $\delta$ within a longer time interval because at some events small values for $w$ and $\delta$ are required, whereas at other events high values for $w$ or $\delta$ are required to get the maximum information content from the data.

Therefore, Peters et al. (2014) suggested the so-called AWAT (Adaptive Window Adaptive Threshold) filter. The innovation in the AWAT-filter consists in the variability of $w$ and $\delta$, which are adjusted according to the

characteristics of the measured data. If the signal strength is high (e.g. due to precipitation), $w$ gets small and if signal strength is low $w$ gets large. Similarly, if noise is high, $\delta$ gets large and if it is low, $\delta$ gets small. The AWAT filter was successfully applied in recent studies (Gebler, et al., 2015; Hannes et al., 2015; Hoffmann et al., 2016).

The threshold approach makes sure that significant weight changes are separated from insignificant changes and leads to a step like course of the calculated cumulative upper boundary flux (see Fig. 6 in Schrader et al. (2013)

or Figs. 6 and 7 in Peters et al. (2014)). The points in time at which the steps occur can be called anchor points and all other points are mere interpolated data.

ET and P are given as the first derivative of cumulative upper boundary flux and are commonly required as the mean for a certain time interval. Since the time span between two anchor points is usually much smaller than one




day, the step interpolation scheme gives fairly good results if only daily resolution is required. However, if the required time interval for the upper boundary flux is much smaller than the time span between the anchor points (e.g. 1 hour or even 10 minutes), the step interpolation yields unrealistic values: At time intervals between two subsequent anchor points the calculated flux is zero. If a time interval comprises one anchor point, the calculated flux is large. Moreover, the magnitude of the flux depends on the length of the chosen time interval since the step occurs immediately. Using such data will probably lead to erroneous simulations and also to numerical problems due to abrupt changes in the boundary conditions with high fluxes alternating with no fluxes.

Note that the step scheme with the abrupt changes directly reflects the resolution of the system. If no further assumptions on the underlying process are justified, this is the maximum information, which can be derived from the measuring setup. Yet, many flux processes at the interface between the soil-plant system and the atmosphere, such as ET or dew fall, are known to be rather smooth and continuous than abrupt.

The aim of this contribution is (i) to show the impact of the step interpolation scheme on calculated fluxes for different time intervals and (ii) to improve the AWAT filter by eliminating the above mentioned problems using linear or cubic Hermitian spline interpolation schemes between the anchor points. This leads to a smoothing of the steps but guarantees that the cumulated fluxes are still exactly the same as in the original approach.

## 2 Material and Methods

### 2.1 Lysimeter setup

The measurements were conducted at the lysimeter station Berlin Marienfelde (52.396731N, 13.367524E). The lysimeter was a so-called grass-reference lysimeter with simulated groundwater depth at 1.3 m. It was 1.5 m deep with a surface area of 1 m². A lever-arm counterbalance system was combined with a laboratory scale, which resulted in an overall resolution of the system of 100 g, which corresponds to approximately 0.1 mm for the upper boundary fluxes. The outflow/inflow of water at the lower boundary was directly recorded with a scale with a resolution of 5 g. The data were logged in a one minute time interval.

The soil material was a packed silt loam taken from a Haplic Phaeozem, which assures good capillary connection between groundwater level and root system. The 20 cm bottom layer consisted of fully water saturated gravel. The 12 cm high grass on the lysimeters was a mixture of *Lolium perenne*, *Festuca arundinacea* and *Poa pratensis*, three cool-season grass species with large rooting depths.

### 2.2 Data processing

The data for this study were recorded from 1 January to 5 August 2014 (Fig. 1). In the time between 2 and 8 April no data were available due to malfunction of the lysimeter scale. In order to evaluate the interpolation schemes, we focussed on three time intervals: (i) 16 to 17 February 2014, representing very low evaporation rates, (ii) 30 to




31 Mai 2014, representing high evaporation rates, and (iii) 07 July 2014 between 13:30 and 15:30, representing the start of a heavy rainfall event.

### 2.3 Threshold and interpolation schemes

The complete filter scheme is given in detail in Peters et al. (2014) and is therefore not explained here. The filter
was applied using a minimum window with of 1 min, a maximum window width of 31 min, a minimum threshold value of 0.1 mm, and a maximum threshold value of 0.24 mm.

#### 2.3.1 Step interpolation scheme

After the moving average (MA) is calculated, the threshold routine distinguishes between significant and insignificant mass changes starting with the first value of the MA at $t = 0$, which might be called the first anchor
point $ap_0$. This value is kept for all subsequent time steps until the difference between the corresponding value of the MA and the anchor point $ap_0$ is greater than the threshold value $\delta$. Then, the new value is the next anchor point $ap_1$ (see Fig. 2 for illustration). This leads to a stepwise course of the calculated cumulative upper boundary flux.

All values between the anchor points can be regarded as interpolated values, whereas the anchor points coincide
exactly with the MA. This procedure guarantees that small oscillations, which occur even after smoothing the data, will not be regarded as real mass changes and thus interpreted as evapotranspiration or precipitation.

#### 2.3.2 Linear and spline interpolation schemes

In order to prevent the above discussed problems, which arise from the step scheme for the upper boundary flux, alternative interpolation schemes can be used. The simplest way is to calculate a linear interpolation between two
subsequent anchor points. An alternative is the use of piecewise Hermitian splines (Fritsch and Carlson, 1980), which smooth the time course of the upper flux but do not oscillate like simple splines. Cubic Hermitian splines are frequently used in soil hydrology, e.g. for the description of hydraulic functions (Iden and Durner, 2007) or for temporal interpolation of measured values in evaporation experiments (Peters and Durner, 2008; Peters et al., 2015). In contrast to the linear interpolation scheme, the spline interpolation yields a smooth curve at the anchor
points and is thus even continuously differentiable.

Such interpolation schemes reflect smooth processes with small changes in small time intervals like evapotranspiration. However, for abrupt changes like rain events, such an interpolation might smooth the data too much and thus lead to unrealistic results again. If, for example, a heavy rain event occurs directly after a longer time with neither evapotranspiration nor precipitation, two subsequent anchor points might comprise a long
time interval and have very different mass values. Then, the new interpolation schemes would yield a low rain intensity for a prolonged time instead of no flux in most of the time interval and a strong rain at the end. This





problem is solved by allowing the above outlined interpolations only for mass decreases (evapotranspiration) or if the mass increase from one to the other anchor point is less than a defined value, e.g. $a\delta$, where $a$ must be greater than one. The latter allows very small precipitation events like dew fall to be smoothed as well. Thus, the step interpolation between two anchor points is kept only if the mass change $\Delta M > a\delta$, which comprises all sorts

of medium to strong precipitation events. We refer to $\delta$ when selecting this scheme because $\delta$ defines the resolution of the system so that mass changes larger than $\delta$ between two anchor points indicate strong signals, which are typical for precipitation events. The parameter $a$ must be larger than 1 but should not be too large to prevent that medium precipitation is smoothed unfavourably. We chose $a = 1.1$ heuristically, meaning that the mass difference must be at least 10 % larger than the system resolution at the specific time.

The linear interpolation scheme as well as the cubic Hermitian Spline interpolation routine of Fritsch and Carlson (1980) were implemented in the AWAT code (Peters et al., 2014). In this study all three interpolation schemes (steps, linear, splines) with $a = 1.1$ for the linear and spline interpolations are applied and compared. In order to test the importance of the rain correction, we additionally applied the linear and spline interpolation schemes without rain correction setting $a$ to a very high value (linear*, spline*).

The fluxes were calculated for time intervals of 1 day, 1 hour, and 10 minutes. The calculated evapotranspiration rates for the three different schemes and time intervals were then compared for the two time spans at 16 to 17 February 2014 and 30 to 31 Mai 2014. The performance of the different schemes, including linear*, spline*, with respect to precipitation following a time with no fluxes was compared for the time span at 07 July 2014 between 13:30 and 15:30. Finally, the biases of the different schemes were compared for the complete data set by

analysing the residuals between filtered and measured data.

### 2.3.3 Definition of bias term

The time series of observations ($O$) can be decomposed as signal and noise:

$$O = R + N \tag{1}$$

where $R$ are the unknown real values and $N$ is the noise. Then the filtered and interpolated time series $F$ is given

by:

$$F(\mathrm{MA}(O)) = F(\mathrm{MA}(R + N)) \tag{2}$$

where MA is the moving average time series. By definition the bias of $F$ ($b_F$) is:

$$b_F := E(F(\mathrm{MA})) - E(R) \tag{3}$$

where $E$ is the linear expected value operator. Considering Eq. [1] yields:

$$b_F = E(F(\mathrm{MA})) - E(O) - E(N) \tag{4}$$

Note that the bias of the first filter step (MA) is given by:





$$b_{\mathrm{MA}} = E(\mathrm{MA}) - E(O) - E(N) \qquad\qquad [5]$$

If we assume $E(N) = 0$ and $E(\mathrm{MA}) - E(O) = 0$ leads to $b_{\mathrm{MA}} = 0$ and

$$b_{\mathrm{F}} = E(F(\mathrm{MA})) - E(O) \qquad\qquad [6]$$

$E(N) = 0$ means that wind and other disturbing factors do not have any significant systematic effects, and $E(\mathrm{MA}) - E(O) = 0$ means that the MA does not lead to systematic deviations between smoothed data and observations. The latter is only given for (i) very small signals, i.e. if the real values ($R$) in the time window $w$ are very similar, or (ii) if $w$ is small, which is the case for the AWAT filter when signals are strong. Thus these assumptions are reasonable and allow to use the distribution of residuals between the mere MA and raw data as reference for the distribution of residuals between interpolated data and raw data.

### 3 Results

#### 3.1 Effect on temporal course of cumulative upper flux

Figure 2 shows the raw data together with the original filter scheme (step) as well as the results of the two other interpolation schemes (linear, spline) for two days with low (left) and high (right) evapotranspiration rates. At the two days in February, the evapotranspiration rates were only approximately 0.35 mm d$^{-1}$, whereas the rates were approximately 5 mm d$^{-1}$ per day at the end of May. By definition, the anchor points coincide with the MA, whereas the step interpolation of the original routine leads to larger differences between interpolated and MA smoothed values. The differences increase with increasing time between two anchor points and with increasing time from the last anchor point. Moreover, this interpolation scheme leads to single, very high changes at the steps and no fluxes during the other time periods, which is especially problematic at low evapotranspiration rates, e.g. at night (see step in upper subplot in Fig. 2, right) or in winter (Fig. 2, left).

Both the linear and spline interpolations lead to smoothed cumulative fluxes, closer to the MA values (Fig. 2). The differences between linear and spline interpolated cumulative fluxes are only minimal except that the spline interpolation leads to slightly more smoothing. The different schemes will have an influence on calculated fluxes for small time intervals as will be shown next.

#### 3.2 Effect on calculated fluxes with different temporal resolution

#### 3.2.1 One day versus one hour intervals

If the required temporal resolution is only one day, the original AWAT filter routine with step interpolation yields sufficient results, since the time intervals between two anchor points are much smaller than one day. The





resulting evapotranspiration rates are shown as grey bars in Fig. 3. However, if the required resolution is one hour, the original step interpolation scheme yields very unrealistic fluxes, especially if potential ET is low (e.g. during night time, or in winter). If a step occurs within an interval, the calculated flux is high, otherwise the flux is zero (Fig. 3, top). The calculated ET reaches a maximum of 15 mm d$^{-1}$ in May and approximately 2.5 mm d$^{-1}$ in

February.

The linear (Fig. 3, center) or spline (Fig. 3, bottom) interpolation schemes lead to smooth and more realistic evapotranspiration prediction. During day time both schemes yield comparable results. However, during night time, the linear scheme predicts small constant ET between two anchor points, whereas the spline scheme predicts a decreasing course until the inflection point between two anchor points is reached, followed by

increasing ET again.

### 3.2.2 10 minute intervals

The unrealistic prediction of ET with the original scheme is even more pronounced if the required time interval gets smaller. For an interval of 10 minutes, the calculated ET can get as high as 35 mm d$^{-1}$ in May and still 15 mm d$^{-1}$ in February or even zero during day time in May (Fig. 4, top). Thus, the fluxes occur not only erratic but

the magnitude of the fluxes within one time interval depends on the selected time interval. This is avoided by the linear or spline interpolation schemes, where the maximum fluxes have roughly the same magnitude for either one hour or 10 minutes intervals (Figs. 3 and 4, center and bottom). Thus, the proposed interpolation schemes allow a more realistic simulation with very high temporal resolution of upper boundary fluxes using lysimeter data, which is important for many physically based studies. Moreover, since precipitation might occur suddenly with

very high fluxes in very short time intervals, selecting such small intervals is important for many simulation studies regarding a realistic expression of precipitation. Only with the new interpolation schemes, such precipitation events can be described in combination with evapotranspiration events within the same temporal resolution.

### 3.3 Analyzing residuals

Figure 5 shows the frequency distribution of the residuals between filtered and measured data. The blue bars show the residuals for the case without threshold value, i.e. for the sole MA and are thus the same for all three compared schemes. These residuals are symmetrically distributed with zero mean, which is expected from a moving average with relatively small window widths, ranging from 1 to 31 minutes. Thus, if the raw data are regarded to be unbiased, the MA can also be regarded as unbiased.

Applying the original step interpolation scheme (Fig. 5, left, red bars) yields a bias towards negative values with a mean of −0.035 mm. This tendency towards negative values is explained by the fact that this interpolation scheme sticks to the mass values at the old anchor points until the threshold is reached, leading to



overestimations of precipitation and underestimations of evapotranspiration periods, with the latter exceeding the former (Peters et al., 2014). Note that applying filters with fixed $w$ and $\delta$ yield even greater biases (see Fig. 8 in Peters et al., 2014).

The simple linear interpolation scheme (Fig. 5, center) leads to a more than 3-fold smaller bias of 0.01 mm with a
slight tendency towards positive values. The spline scheme (right) leads even to a slightly smaller deviation. Thus, the linear and spline interpolation schemes are not only superior for the selected time spans in February and May but also for the complete measured period. The additional computational burden is only minor for any interpolation scheme in comparison with the preceding AWAT filtering. Thus, we suggest to always use the spline scheme.

**3.4 Effect on rain events**

If a relatively strong precipitation event follows a prolonged period with no significant flux, the mere interpolation schemes without rain correction smooth such an event in an unrealistic manner (linear* and spline* in Fig. 6). The heuristic selection criterion determines, that the step interpolation is kept for time intervals between two anchor points if $\Delta M > 1.1\,\delta$ (linear and spline). This prevents unfavourable smoothing at the beginning of rain events.

**4 Summary and Conclusions**

The original step interpolation scheme of the threshold routine of the AWAT yields unrealistic fluxes with abrupt changes for short time intervals. This is most pronounced when real fluxes are small and therefore the distance between two anchor points is in the same magnitude or larger than the chosen time interval. This is problematic if highly resolved boundary conditions are needed for e.g. physically based simulations of water and energy fluxes
in the soil-plant atmosphere system.

Improving the filter by the proposed interpolation schemes solves this problem leading to smoothed values, which are more realistic, especially for evapotranspiration events. Moreover, the spline scheme allows even a continuous differentiation and thus any temporal resolution for the predicted fluxes. A simple heuristic selection criterion, which separates medium to strong precipitation from all other events, prevents that such precipitations
are smoothed in an unfavourable way. Thus, upper boundary conditions for physically based simulations with very short time intervals can now be automatically derived from precision lysimeters.

In this study, we used a counterbalance weighing system with approximately 0.1 mm resolution. Modern lysimeters resting on weighing cells (von Unold and Fank, 2008) can have a resolution up to 0.01 mm. Then, the problems of the step interpolation scheme is less pronounced but still present, specifically at times with low
fluxes. Thus, the proposed solution is important especially for lysimeters with limited resolution, which are still often used, but is also favourable for systems with higher resolution.



Note that the results and conclusions regarding the interpolation schemes hold also for filters with fixed window widths and threshold values (e.g. Fank, 2013; Schrader et al., 2013).

**Acknowledgements**

This study was financially supported by the Deutsche Forschungsgemeinschaft (DFG grant PE 1912/2-1). We thank Michael Facklam, Reinhild Schwartengräber, Björn Kluge, Joachim Buchholz and Steffen Trinks for their assistance with the lysimeter construction and maintenance. We also thank Marnik Vanclooster as Associate Editor for his insightful comments and suggestions, which greatly improved the manuscript.

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





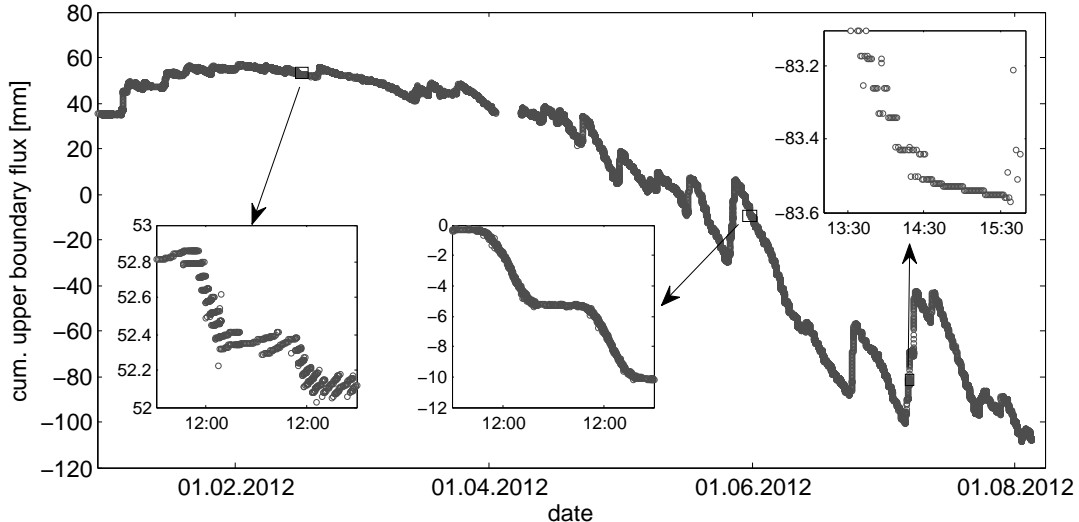

**Fig. 1.** Raw data for cumulative upper boundary flux of a grass covered lysimeter in Berlin-Marienfelde, Germany. The data of the three selected time intervals at 16 to 17 February 2014; 30 to 31 Mai 2014, and 07 July 2014 between 13:30 and 15:30 are given in the three subplots. Note that the time and flux intervals for the three intervals are different in the subplots.





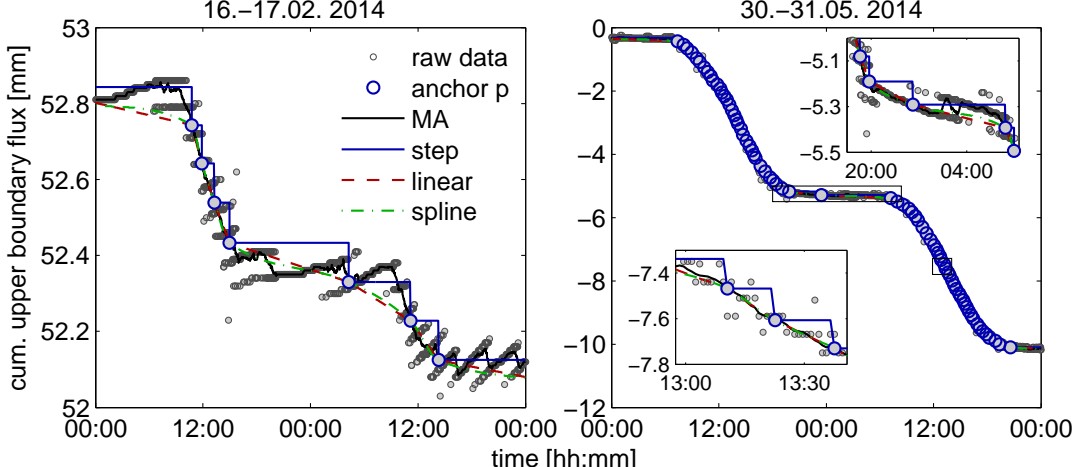

**Fig. 2.** Raw data of two evapotranspiration events, filtered with original AWAT filter (steps) and linear as well as spline interpolation schemes. Left: low evapotranspiration at 16 to 17 February 2014; right: high evapotranspiration rates at 30 to 31 Mai 2014.





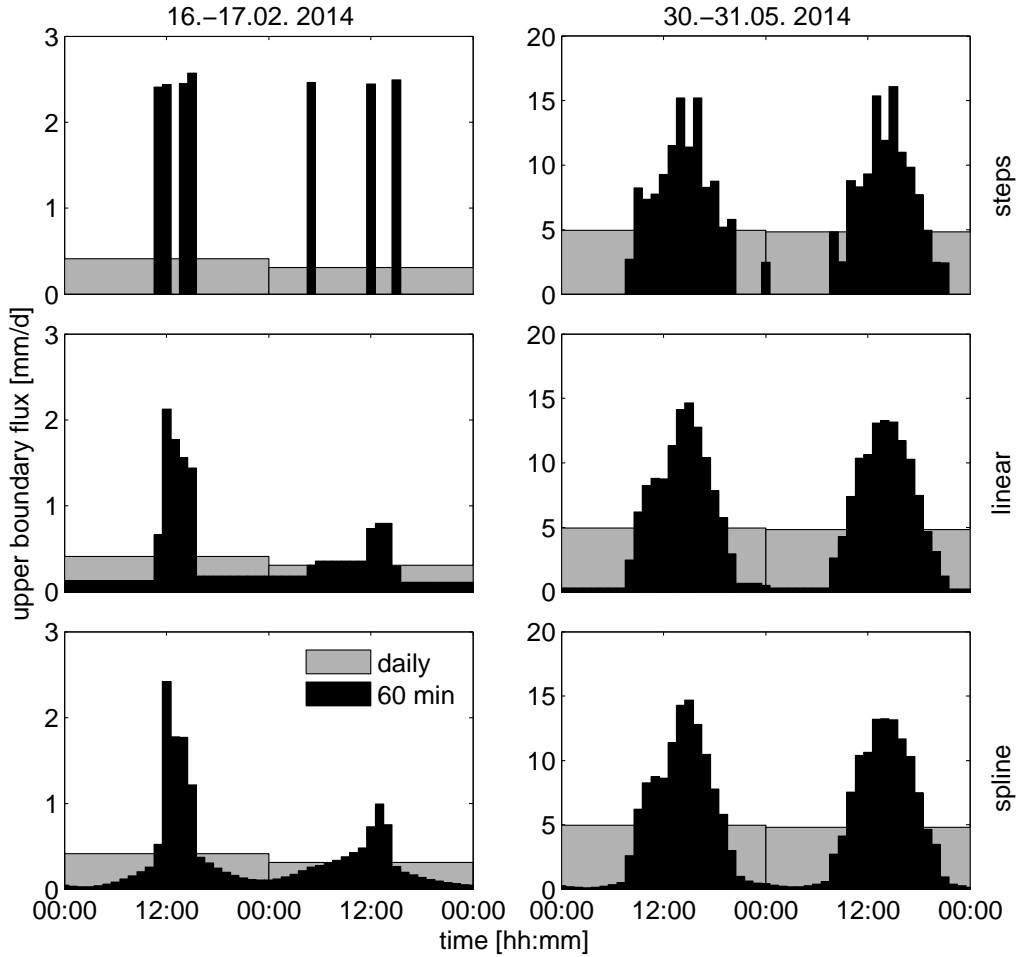

**Fig. 3.** Derived evapotranspiration rates from data shown in Fig. 2 with temporal resolution of one day or one hour, respectively. Steps: original step interpolation scheme; linear: linear interpolation scheme; spline: cubic Hermitian spline interpolation scheme.





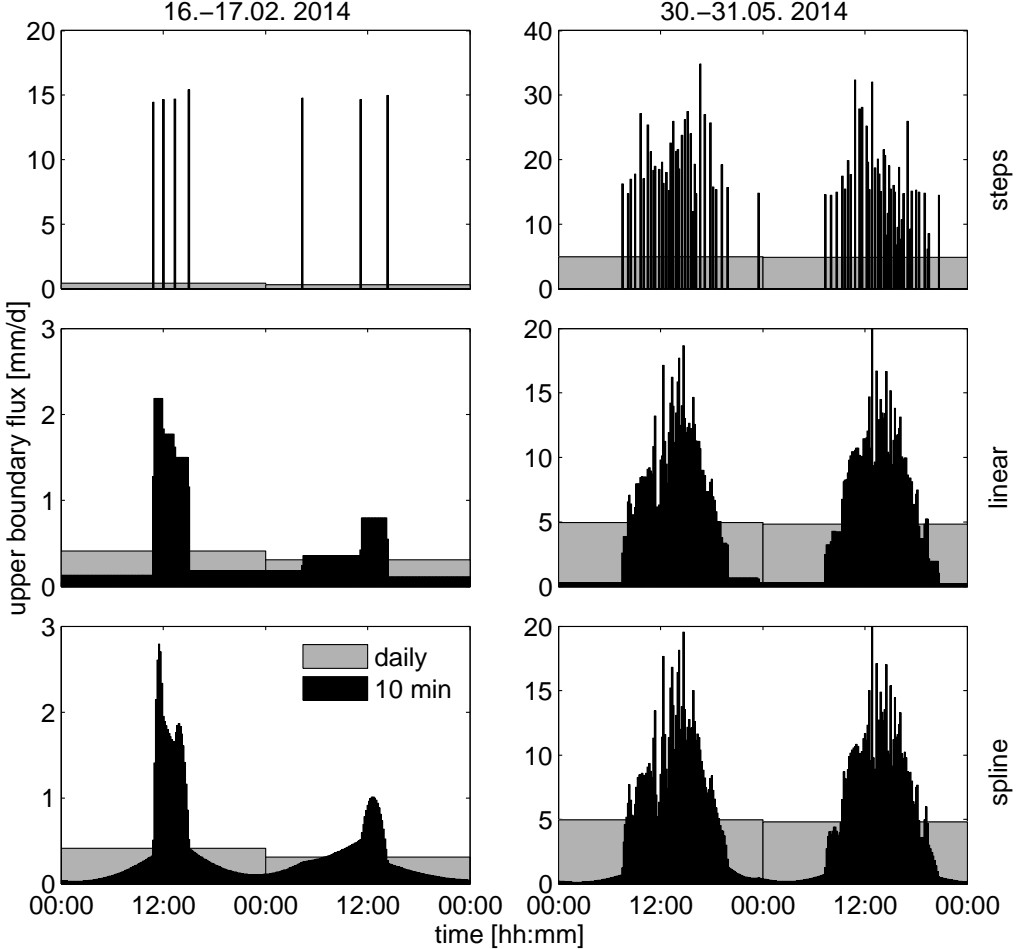

**Fig. 4**. Derived potential evapotranspiration rates from data shown in Fig. 2 with temporal resolution of one day or 10 minutes, respectively. Steps: original step interpolation scheme; linear: linear interpolation scheme; spline: cubic Hermitian spline interpolation scheme. Note different scales on ordinates for the step scheme between
5   Figs. 4 and 3.





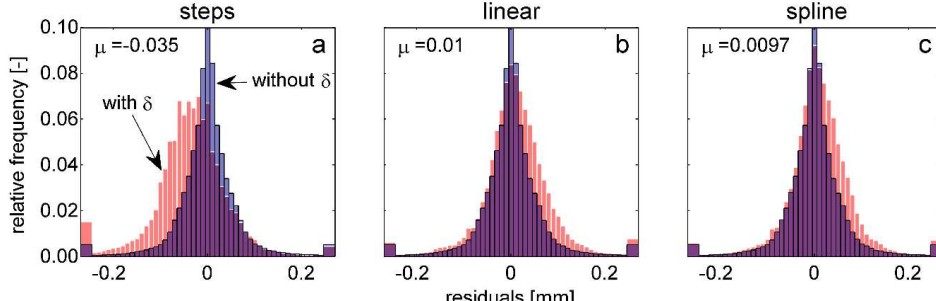

**Fig. 5.** Relative residual frequency distribution for the complete data set and the different interpolation schemes. Blue bars indicate residuals between original and filtered data for the cases with mere smoothing, omitting the threshold values; red bars indicate cases with threshold value and subsequent interpolation. The broad bars at plot edges comprise all residuals greater than 0.25 or smaller than −0.25 mm. Steps: original step interpolation scheme; linear: linear interpolation scheme; spline: cubic Hermitian spline interpolation scheme.




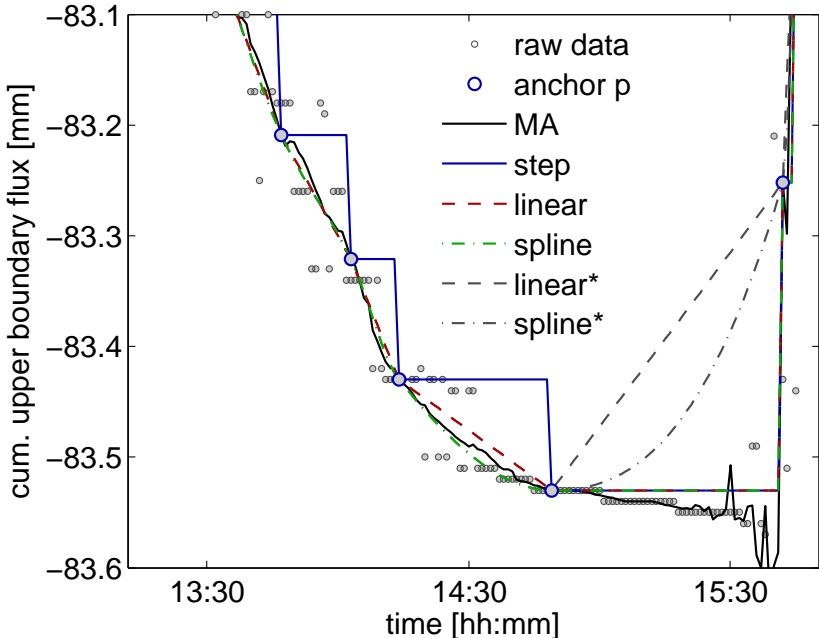

**Fig. 6.** Raw data of a period of evaporation followed by a precipitation event at 7 July 2014. Anchor p: anchor point; MA: moving average; steps: original step interpolation scheme; linear: linear interpolation scheme; spline: cubic Hermitian spline interpolation scheme; linear*: linear interpolation scheme without precipitation correction; spline*: cubic Hermitian spline interpolation scheme without precipitation correction.

