# Peer review of "Technical note: Improving the AWAT filter with interpolation schemes for advanced processing of high resolution data"

_Hydrology and Earth System Sciences, 2016_

## Referee Comment (RC1) · Anonymous Referee #1 · 9 Apr 2016

This paper aims at improving the automatic processing of time series from high resolution-lysimeters, allowing one to better estimate the evapotranspiration and rainfall effects. In a scientific context evolving towards data-mining processes, such an investigation is very useful and deserves publication as a technical note in HESS, when the following comments are taken into account.

Detailed comments: 1. P2 L20-23: need not to be discussed in this paper, but suggestion for future work: have you looked at routines used to process GNSS (GPS, GLONASS) time series, where identifying steps is challenging as well? 2. P2 L25: "if the signal strength is high" . . ."noise is high": could you quantify? 3. P3 L4: the flux is zero: but what happens if the distance between anchors is reduced? 4. P5 l14: what

is a "very high value"? 5. P5 L18: "with no fluxes": I suppose that ET plays a major role in July. So I don't understand "no flux". 6. P5 L24: which filter? As described in 2.3? Elaborate. 7. P6 L20: "problematic": I do not understand your point. As even at night a small slope (probably significant, this may be tested) appears, this implies steps. So, what's the problem? The fact that an apparently smooth decrease in fluxes appear as an abrupt change when looking at steps? Why is it quantified as "high changes"? Incidentally, why do the raw data on Fig 2 (left) appear as sawtooth, i.e. as small groups, of increasing slopes (and to a lesser extent, in an opposite way on fig 2 right, upper panel), while on Figure 6 the raw data are rather grouped by constant levels? Elaborate. Minor details, typos... 8. P2 L32: derivative of the cumulative... 9. P2L32-33: the syntax of this sentence ("ET...interval") looks strange. Elaborate. What is "certain"? 10. p3 L29: "in the time between": prefer: ""between 2 and 8 April, no data..." 11. P6 L14: "At the two days": prefer: "On February 16 and 17, ..." 12. P6 l15: "only approximately": prefer: "the ET rate is estimated at the XX level" (and if you can provide an error bar, just add it). 13. P6 L23: "are only minimal": what do you mean? "the difference is negligible"? 14. P8 L18: "in the same magnitude": I suppose: "is similar of larger...". 15. P12: mai –> May

Please also note the supplement to this comment:
http://www.hydrol-earth-syst-sci-discuss.net/hess-2016-51/hess-2016-51-RC1-supplement.pdf

---

## Referee Comment (RC2) · J. Fank (Referee) · 9 Apr 2016

General comments

The technical note show that the step interpolation scheme used in the AWAT filter, which reflects the resolution of the measuring system, can lead to unrealistic prediction of P and ET, if they are required in high temporal resolution (hourly or shorter time steps). Linear and spline interpolation schemes are introduced to overcome these problems. The presented methods are very useful in estimating precise values for P and ET from weighing lysimeter measurements with a high mass and temporal resolution if the diurnal course of P and ET must be known, e.g. if root water uptake processes shall be simulated using physically based models, or macro pore flow and

solute transport due to heavy but short precipitation events shall be simulated under realistic conditions.

Specific comments

In the AWAT filter the delta-value is set to the resolution of the measuring system, which leads to a step interpolation scheme. That means that values given below the resolution of the measuring system are random and are not allowed to be interpreted as measured data. Therefore in my opinion the methods presented in the paper are not part of the data evaluation process but at the starting point of data interpretation. Although the presented improvement of the AWAT filtering method is of very high importance for further interpretation of lysimeter data and of their use in process oriented numerical modeling, I suggest the authors to remark on the point where data evaluation ends and data interpretation is going to start.

Technical corrections

P2 L32: . . .derivative of the cumulative. . .

P3 L29: Between 2 and 8 April, . . .

P6 L14: On February 16 and 17, the evapotranspiration rates were only approximately 0.35 mm d-1, whereas the ET rates were estimated at the 5 mm d-1 level at the end of May.

P6 L23: . . . cumulative fluxes are negligible except that the . . .

P11 Fig. 1: Please check, if the presented window for the starting point of a rainfall event (07 July 2014 13:30 to 15:30) is at the correct position in the graph of the cum. upper boundary flux.

P12 Fig. 2: Mai –> May

Please also note the supplement to this comment:

[Figure]

http://www.hydrol-earth-syst-sci-discuss.net/hess-2016-51/hess-2016-51-RC2-supplement.pdf

---

## Author Comment (AC1) · 13 Apr 2016

Dear Reviewer,

we are very thankful for the generally positive judgement of our manuscript and for the specific comments, which help us to strengthen the paper.

Please find below all replies to the comments as inserted blue text.

Kind regards,

Andre Peters, Thomas Nehls and Gerd Wessolek

This paper aims at improving the automatic processing of time series from high resolution-lysimeters, allowing one to better estimate the evapotranspiration and rainfall effects. In a scientific context evolving towards data-mining processes, such an investigation is very useful and deserves publication as a technical note in HESS, when the following comments are taken into account.

Detailed comments:

1. P2 L20-23: need not to be discussed in this paper, but suggestion for future work: have you looked at routines used to process GNSS (GPS, GLONASS) time series, where identifying steps is challenging as well?

No but we are thankful for that hint and will come back to it if we further improve the filter routine. However, we want to emphasize here that this contribution is not meant to help identifying the steps but to avoid them when interpreting the data.

2. P2 L25: "if the signal strength is high" ..."noise is high": could you quantify?

This is no easily done in the text. The signals can be extremely high if strong precipitation like a rain storm event takes place (several mm in a few minutes; Fig. 1 in the original paper of Peters et al., 2014; heavy precipitation event). Noise made up to almost 2 mm fluctuations without a significant signal (see strong wind event in the same Fig.). Yet, under other climatic conditions both can be even higher. As this part of the introduction is meant to be very general, we would like to prefer not discussing this issue in depth here.

3. P3 L4: the flux is zero: but what happens if the distance between anchors is reduced?

The flux (first derivative of cumulative flux with respect to time) is zero between two anchor points by definition in the case of the step scheme, irrespectively of the distance between anchor points or the magnitude of delta (see Fig. 2 for instance). Steps mean that the calculated flux is either zero or, at the step, very high. We think that this is clear from the text and the figures in both the manuscript and the original paper (Peters et al., 2014).

4. P5 l14: what is a "very high value"?

We set it arbitrarily to 9999, which means that no rain correction is made since no step is higher than 9999 delta. This is now added by modifying the sentence to "In order to test the importance of the rain correction, we additionally applied the linear and spline interpolation schemes without rain correction setting $a$ to the very high value of 9999 (linear*, spline*). This guaranteed that the criterion $\Delta M > a\delta$ is never met."

5. P5 L18: "with no fluxes": I suppose that ET plays a major role in July. So I don't understand "no flux".

Before the rain event started, ET became less as shown in Fig. 6. However, ET was not zero, thus we will change the sentence to "… with low flux…". We thank the reviewer for this hint.

6. P5 L24: which filter? As described in 2.3? Elaborate.

Yes, this part is derived for the general filtering scheme with using first the MA and then the threshold filtering with interpolation. In order to make it clearer we introduced "…(as described above)…"

7. P6 L20: "problematic": I do not understand your point. As even at night a small slope (probably significant, this may be tested) appears, this implies steps. So, what's the problem? The fact that an apparently smooth decrease in fluxes appear as an abrupt change when looking at steps? Why is it quantified as "high changes"?

This is the central point of the manuscript: Each step for ET calculation is somehow problematic since ET does not occur in steps but rather continuously. Since the magnitude of the step is at least $\delta_{min}$ this is especially problematic for low "real" ET fluxes since then a continuously small ET within several hours is lumped into one single step as shown in Fig. 2. This is now written clearer by modifying the sentence to: "Moreover, this interpolation scheme leads to single, very high changes at the steps and no fluxes during the other time periods, which is especially problematic at low evapotranspiration rates, e.g. at night (see step in upper subplot in Fig. 2, right) or in winter (Fig. 2, left), where the continuously low ET fluxes of several hours are lumped into one single step."

Incidentally, why do the raw data on Fig 2 (left) appear as sawtooth, i.e. as small groups, of increasing slopes (and to a lesser extent, in an opposite way on fig 2 right, upper panel), while on Figure 6 the raw data are rather grouped by constant levels?  Elaborate.

We thank the reviewer for this question, which needs to be answered in the manuscript. The sawtooth shape is caused by the measurement system consisting of two scales with different resolution. We add a small paragraph at the end of section 2.2:

"Note that the "sawtooth" shape of the first subplot is caused by the two scales with different resolution. If outflow at the lower boundary occurs, each 5 g outflow is recorded in the data leading to an apparent increase of cumulative outflow. If approximately 100 g flew out, the lysimeter scale records an apparent decrease of cumulative outflow of 100 g. This is repeated and sometimes superimposed by a real signal like ET or P."

Minor details, typos

8. P2 L32: derivative of the cumulative...

Has been changed

9. P2 L32-33: the syntax of this sentence ("ET....interval") looks strange.

We do not understand. To our knowledge the sentence is correct and can be understood. Thus, we would like to keep it as it is except the minor modification given in point 8.

Elaborate. What is "certain"?

This is explained in the two sentences following this sentence. However, we substituted "certain" by "application specific"

10.p 3 L29: "in the time between": prefer: ""between 2 and 8 April, no data..."

Thanks, has been changed

11. P6 L14: "At the two days": prefer: "On February 16 and 17, ..."

Thanks, has been changed

12. P6 l15: "only approximately": prefer: "the ET rate is estimated at the XX level" (and if you can provide an error bar, just add it).

No, ET is not estimated but can be approximately derived from visual inspection of Fig. 2 if we subtract the cumulative fluxes between two night times. Error bars cannot be given since only two days are given in each subplot. We just omitted the word "only" to make the sentence clearer.

13. P6 L23: "are only minimal": what do you mean? "the difference is negligible"?

Yes, has been changed

14. P8 L18: "in the same magnitude": I suppose: "is similar of larger...".

Has been changed

15. P12: mai → May

Thank you very much, has been changed

---

## Author Comment (AC2) · 13 Apr 2016

Dear Johann Fank,

we are very thankful for your generally positive judgement of our manuscript and for the specific comments, which help us to strengthen the paper.

Please find below all replies to the comments as inserted blue text.

Kind regards,

Andre Peters, Thomas Nehls and Gerd Wessolek

General comments

The technical note show that the step interpolation scheme used in the AWAT filter, which reflects the resolution of the measuring system, can lead to unrealistic prediction of P and ET, if they are required in high temporal resolution (hourly or shorter time steps). Linear and spline interpolation schemes are introduced to overcome these problems. The presented methods are very useful in estimating precise values for P and ET from weighing lysimeter measurements with a high mass and temporal resolution if the diurnal course of P and ET must be known, e.g. if root water uptake processes shall be simulated using physically based models, or macro pore flow and solute transport due to heavy but short precipitation events shall be simulated under realistic conditions.

Specific comments

In the AWAT filter the delta-value is set to the resolution of the measuring system, which leads to a step interpolation scheme. That means that values given below the resolution of the measuring system are random and are not allowed to be interpreted as measured data. Therefore in my opinion the methods presented in the paper are not part of the data evaluation process but at the starting point of data interpretation. Although the presented improvement of the AWAT filtering method is of very high importance for further interpretation of lysimeter data and of their use in process oriented numerical modeling, I suggest the authors to remark on the point where data evaluation ends and data interpretation is going to start.

We are very thankful for this comment. This is exactly what we wanted to state in the sentence "Note that the step scheme with the abrupt changes directly reflects the resolution of the system. If no further assumptions on the underlying process are justified, this is the maximum information, which can be derived from the measuring setup." In the introduction section. We added now the sentence "As stated above, the step interpolation scheme directly reflects the resolution of the measurement system and is therefore the final part of a mere data evaluation process. Using the suggested two interpolation schemes is the first step towards data interpretation." (section 2.3.2).

Technical corrections

P2 L32: derivative of the cumulative

Thanks, has been changed

P3 L29: Between 2 and 8 April,

Thanks, has been changed

P6 L14: On February 16 and 17, the evapotranspiration rates were only approximately 0.35 mm d-1, whereas the ET rates were estimated at the 5 mm d-1 level at the end of May.

Thanks, has been changed

P6 L23: cumulative fluxes are negligible except that the

Thanks, has been changed

P11 Fig. 1: Please check, if the presented window for the starting point of a rainfall event (07 July 2014 13:30 to 15:30) is at the correct position in the graph of the cum. upper boundary flux.

The box is at the correct position. At those days there were several rain events with small interruptions with evapotranspiration taking place.

P12 Fig. 2: Mai –> May

Thanks, has been changed

---

## Referee Comment (RC3) · T. Puetz (Referee) · 10 May 2016

General comments Filter procedures for lysimeter data are necessary tools to process the data records. The AWAT filter can be used as a useful / timesaving tool for data preparation. In my understanding, a filter must find only improper, incorrect, or faulty data in order to correct these errors in the next step. Within very narrow limits, an evaluation of the data is necessary to classify their sense and correctness. However, an interpretation of the data is strictly to avoid.

Specific comments In your introduction: beside P and ET you should mention the importance of the seepage water because of the importance for the water balance. P2 L 4-5: here I miss also the seepage water or drainage!! P 5 L 18 "a time with no fluxes

was compared". - It is hard to believe that there is no flux (= no ET) in July? You did not discuss or reflect to any data noise induced by wind events. Are you sure to have no wind effects? For further filter tests, a combination of different, changing scenarios would be desirable, e.g. a mixed scenario of rain – ET – rain? Why no synthetic data were used, because for this case very specific data mistakes can be inserted? While real lysimeter data always an interpretation must be carried out to define the true values.

Technical corrections I will list only errors that have not been criticized by the former reviewer. P 8 L 23: What is a simple heuristic selection criterion? P 11 Fig 1: the legend of the x-axis and date below are showing different years 2012 / 2014 than in the description? P 13 Fig 3: this figure is not a really good graphic to compare results, my suggestion: compare it as differences P 14 Fig 4: see above!

Please also note the supplement to this comment:
http://www.hydrol-earth-syst-sci-discuss.net/hess-2016-51/hess-2016-51-RC3-supplement.pdf

---

## Author Comment (AC3) · 20 May 2016

Dear Thomas Pütz,

we are very thankful for your generally positive judgement of our manuscript and for the specific comments, which helped us to strengthen the paper.

Please find below all replies to the comments as inserted blue text.

Kind regards,

Andre Peters, Thomas Nehls and Gerd Wessolek

General comments

Filter procedures for lysimeter data are necessary tools to process the data records. The AWAT filter can be used as a useful / timesaving tool for data preparation. In my understanding, a filter must find only improper, incorrect, or faulty data in order to correct these errors in the next step. Within very narrow limits, an evaluation of the data is necessary to classify their sense and correctness. However, an interpretation of the data is strictly to avoid.

Yes and no. We agree that a data filter should primarily help to eliminate faulty data and noise. Yet, if we want to use the final data we must interpret them. This is always a delicate step since it requires expert knowledge. We hope that we showed in the paper that omitting the suggested interpolation schemes and keep the mere step interpolation can lead to a "wrong" data interpretation. This is of particular importance if data shall be used for modeling in high temporal resolution. Since both reviewers, you and Johann Fank, have raised this issue we added now the sentence "As stated above, the step interpolation scheme directly reflects the resolution of the measurement system and is therefore the final part of a mere data evaluation process. Using the suggested two interpolation schemes is the first step towards data interpretation." (section 2.3.2).

Specific comments

In your introduction: beside P and ET you should mention the importance of the seepage water because of the importance for the water balance.

We agree that seepage water is an important part of the water balance. However, this Note deals exclusively with the filtering and interpretation of data for P and ET, which can be directly derived from lysimeter measurements including seepage. Seepage water depends directly on the imposed lower boundary of the lysimeter and might thus not be representative for the place where the lysimeter is located. Since this discussion is beyond the scope, we would like to omit it here. Furthermore, in terms of discussing noise and filtering of noise, seepage is of smaller interest since the noise is much more reduced due to the transport process of water within the lysimeter.

P2 L 4-5: here I miss also the seepage water or drainage!!

See reply above

P 5 L 18 "a time with no fluxes was compared". - It is hard to believe that there is no flux (= no ET) in July?

This is right. We changed it to "…low fluxes…".

You did not discuss or reflect to any data noise induced by wind events. Are you sure to have no wind effects? For further filter tests, a combination of different, changing scenarios would be desirable, e.g. a mixed scenario of rain – ET – rain?

This note is not meant as a test of the AWAT filtering scheme. It is intended as an extension towards data interpretation. Strong wind does only mean that the value for delta is high so that the step and therefore the vertical distance of two consecutive anchor points is large. The wind effects are discussed and handled in the original paper (Peters et al., 2014).

Why no synthetic data were used, because for this case very specific data mistakes can be inserted? While real lysimeter data always an interpretation must be carried out to define the true values.

This Note deals with data after calculation of the anchor points, i.e. after all noise and errors are assumed to be eliminated. For the schemes, which are introduced here, it makes no difference whether real data or synthetic data is used. The use of synthetic data is interesting and might be used in future studies to test the filter throughout. However, as already discussed by Peters et al., (2014) artificially composed data might not comprise the same complex system and noise behavior as in reality.

Technical corrections

I will list only errors that have not been criticized by the former reviewer.

P 8 L 23: What is a simple heuristic selection criterion?

The heuristic selection criterion is introduced in section 2.3.2. What it does is given in the second part of the sentence to which the reviewer refers.

P 11 Fig 1: the legend of the x-axis and date below are showing different years 2012 / 2014 than in the description?

We are thankful for this hint and apologize for the fault. All data was recorded in 2014. This is now corrected.

P 13 Fig 3: this figure is not a really good graphic to compare results, my suggestion: compare it as differences. P 14 Fig 4: see above!

The aim of Fig 3 and 4 is to show that the step interpolation scheme leads to predicted ET, which (i) depend on the chosen time interval for the output and (ii) are either very high or zero and that the suggested interpolation schemes solve this specific problem. This is best done by the figures as they are.